# The Modulation of Respiratory Epithelial Cell Differentiation by the Thickness of an Electrospun Poly-ε-Carprolactone Mesh Mimicking the Basement Membrane

**DOI:** 10.3390/ijms25126650

**Published:** 2024-06-17

**Authors:** Seon Young Choi, Hyun Joo Kim, Soyoung Hwang, Jangho Park, Jungkyu Park, Jin Woo Lee, Kuk Hui Son

**Affiliations:** 1Department of Thoracic and Cardiovascular Surgery, Gachon University Gil Medical Center, College of Medicine, Gachon University, Incheon 21565, Republic of Korea; vet904rainbow@gmail.com (S.Y.C.); applechips@naver.com (H.J.K.); snrntlwy1004@gmail.com (S.H.); 2Department of Molecular Medicine, College of Medicine, Gachon University, Incheon 21999, Republic of Korea; hjp0904@naver.com (J.P.); jungkyu8131@gachon.ac.kr (J.P.); 3Department of Health Sciences and Technology, Gachon Advanced Institute for Health Sciences & Technology (GAIHST), Gachon University, Incheon 21999, Republic of Korea

**Keywords:** membrane thickness, goblet cell hyperplasia, epithelial–mesenchymal transition, PCL nanofiber, airway disease model

## Abstract

The topology of the basement membrane (BM) affects cell physiology and pathology, and BM thickening is associated with various chronic lung diseases. In addition, the topology of commercially available poly (ethylene terephthalate) (PET) membranes, which are used in preclinical in vitro models, differs from that of the human BM, which has a fibrous and elastic structure. In this study, we verified the effect of BM thickness on the differentiation of normal human bronchial epithelial (NHBE) cells. To evaluate whether the thickness of poly-ε-carprolactone (PCL) mesh affects the differentiation of NHBE cells, cells were grown on thin- (6-layer) and thick-layer (80-layer) meshes consisting of electrospun PCL nanofibers using an air–liquid interface (ALI) cell culture system. It was found that the NHBE cells formed a normal pseudostratified epithelium composed of ciliated, goblet, and basal cells on the thin-layer PCL mesh; however, goblet cell hyperplasia was observed on the thick-layer PCL mesh. Differentiated NHBE cells cultured on the thick-layer PCL mesh also demonstrated increased epithelial–mesenchymal transition (EMT) compared to those cultured on the thin-layer PCL mesh. In addition, expression of Sox9, nuclear factor (NF)-κB, and oxidative stress-related markers, which are also associated with goblet cell hyperplasia, was increased in the differentiated NHBE cells cultured on the thick-layer PCL mesh. Thus, the use of thick electrospun PCL mesh led to NHBE cells differentiating into hyperplastic goblet cells via EMT and the oxidative stress-related signaling pathway. Therefore, the topology of the BM, for example, thickness, may affect the differentiation direction of human bronchial epithelial cells.

## 1. Introduction

The natural basement membrane (BM) of the human respiratory epithelium is a very thin (0.1–2 μm) sheet composed of extracellular matrix (ECM) proteins, such as collagen IV, laminin, and proteoglycan, onto which epithelial cells are anchored [1,2]. Topological features of the BM, such as thickness, pore size, and porosity, are associated with the various important roles in cellular processes, including proliferation, migration, and differentiation [3,4]. Moreover, the topological characteristics of the BM affect numerous physiological and pathological processes [5,6,7].

In preclinical in vitro models used to assess the structure and function of the respiratory system, primary cells are grown on three-dimensional (3D) Transwell inserts with a microporous membrane under air–liquid interface (ALI) conditions to generate mucociliary epithelium [8,9]. Respiratory epithelium, properly differentiated from normal human bronchial epithelial (NHBE) cells, has a pseudostratified structure and contains multiple types of cells such as ciliated, goblet, and basal cells, similar to normal human respiratory epithelium. A synthetic poly (ethylene terephthalate) (PET) membrane with 10 μm thickness is often used in these models to create the requisite ALI [10]. However, the PET membrane is relatively stiffer than the natural BM. The elastic (Young’s) modulus of the basement membrane in adult tissues varied in the 1–4 MPa range, which is lower than that of PET (>2900 MPa) [11,12,13]. Furthermore, the Young’s modulus in porcine lung tissue was in the 1.4–6.1 kPa range [14]. Thus, it does not effectively mimic the elasticity of the natural BM [15,16], as thick and abnormally stiff BM has been associated with goblet hyperplasia and chronic lung disease such as asthma, chronic obstructive pulmonary disease (COPD), and idiopathic pulmonary fibrosis (IPF) [17,18,19].

To overcome the limitation of the conventional PET membrane, various electrospun polymers have been examined for their capacity to better mimic the fibrous human BM and airway wall model [20,21]. Among them, PCL is attracting attention due to its flexibility, ease of copolymer production, and long degradation time. PCL is a semi-crystalline polymer (~50%) with a melting point between 59 °C and 64 °C and a glass transition temperature of ~60 °C. It can be synthesized from ring-opening polymerization of linear caprolactone monomers [22,23]. Normally, PCL occurs in bulk degradation in vivo, and its main products are caproic acid, succinic acid, valeric acid, and butyric acid [24]. Compared to other polymers, it has a relatively long degradation time of up to 2–4 years [25,26]. Because caprolactone polymer is one of the most flexible synthetic polymers, it is combined with copolymers to increase the flexibility of the resulting biomaterial. Notably, electrospun PCL is widely documented for its excellent mechanical properties and slow degradation in vivo and has been found to be a suitable mimicker of elastic airway tissue [27,28]. Jain et al. showed that lung epithelial and endothelial cells formed imperfect cellular sheets on PET membranes due to the difference in topology and thickness between the PET and PCL membranes [28]. Although the study suggested the potential of synthetic PCL mesh to mimic the human tissue BM, the effect of electrospun PCL on the generation of mucociliary epithelium remains unclear. Additionally, there is a lack of research on how electrospun PCL affects the differentiation and signaling pathway of primary airway cells.

Herein, we hypothesized that the topology, such as the thickness of the electrospun PCL mesh, affects the differentiation of NHBE cells in an ALI cell-culture model. To test this hypothesis, we compared the differentiation of NHBE cells cultured on electrospun 6-layer (thin) and 80-layer (thick) PCL meshes. Our results showed that the NHBE cells differentiated on the 80-layer PCL mesh generated abnormal respiratory epithelium through an increased number of goblet cells compared to the 6-layer PCL mesh through EMT- and oxidative stress-related signaling pathways.

## 2. Results

### 2.1. Characteristics of Electrospun PCL Mesh

To mimic the BM, two types of mesh (6-layer and 80-layer) were prepared using an electrospun PCL nanofiber. Since electrospinning was performed with different numbers of layers under the same conditions, there was no significant difference in the surface morphology between the 6-layer and 80-layer PCL meshes (Figure 1a,c).

The 6-layer and 80-layer meshes were 18.75 ± 1.92 μm and 256.83 ± 21.64 μm thick, respectively (Figure 1b,d; Table 1). According to the SEM image analysis, the pore size was 0.59 ± 0.22 μm for the 6-layer PCL mesh and 0.64 ± 0.41 μm for the 80-layer mesh. The fiber diameter was 0.23 ± 0.04 μm for the 6-layer mesh and 0.25 ± 0.02 μm for the 80-layer mesh. The pore size and fiber diameters of the two mesh types were not significantly different.

The mechanical properties of the two mesh types were measured using a universal testing machine. The obtained stress–strain curve of the electrospun PCL meshes provides information about their tensile strength and strain under tension (Figure 1e). The elastic modulus of the 6-layer PCL mesh was 18.81 ± 2.20 N/mm, and that of the 80-layer mesh was 19.36 ± 3.22 N/mm (Figure 1f). The ultimate tensile strength of the 6-layer PCL mesh was 9.62 ± 1.20 MPa, and that of the 80-layer mesh was 10.20 ± 1.24 MPa (Figure 1g). The representative properties of the 6- and 80-layer PCL mesh, including the stress–strain curve, elastic modulus, and ultimate tensile strength, were not significantly different.

### 2.2. Viability and Barrier Function of Differentiated NHBE Cells during ALI

As shown in Figure 2a, NHBE cells were found to differentiate when cultured on PCL mesh inserts during the ALI. To assess whether the collagen-coated PCL mesh and PET Transwell inserts had any toxic effects on the NHBE cells, cell viability was measured using the CCK-8 assay. The results showed that the viability of the NHBE cells differentiated on the 6- and 80-layer PCL meshes did not decrease up to 28 days post-initiation of the ALI. However, the viability of the NHBE cells differentiated on the PET Transwell inserts (the control condition) slightly decreased (* *p* < 0.05) at day 28 (Figure 2b).

The barrier function of differentiated NHBE cells cultured on the 6- and 80-layer PCL meshes was assessed by examining by trans-epithelial electrical resistance (TEER) values. The results showed that the TEER values of the NHBE cells differentiated on the 6- and 80-layer PCL meshes were maintained after the initiation of the ALI. Likewise, the TEER values of the NHBE cells differentiated on PET Transwell inserts were also maintained during the ALI (Figure 2c). In particular, the TEER values of the NHBE cells differentiated on the 6- and 80-layer PCL meshes were higher than those of the cells differentiated on the PET Transwell inserts. Based on these results, only cells grown on the 6- and 80-layer PCL meshes were used in the subsequent characterization test conducted on day 21.

### 2.3. Generation of Airway Epithelium on the 6- and 80-Layer PCL Mesh

Differentiated pseudostratified epithelium was present for up to 28 days after the initiation of the ALI. Within seven days of the initiation of the ALI, the epithelial layer on the 6- and 80-layer PCL meshes was at least two to three cells thick (Figure 2d). Histological images showed that the NHBE cells differentiated on the 6-layer PCL mesh generated a two- to three-layer epithelium with well-differentiated cells such as goblet and ciliated cells. However, the epithelial layer of the NHBE cells differentiated on the 80-layer PCL mesh had hyperplastic goblet cells 14 days after the initiation of the ALI. Moreover, there were scant ciliated cells in the epithelial layer on the 80-layer PCL mesh 21 days after the initiation of the ALI (Figure 2d). As the control, the NHBE cells differentiated on the PET Transwell generated epithelium with goblet and ciliated cells (Appendix A).

The levels of specific marker genes in the NHBE cells during the ALI on the 6- and 80-layer PCL meshes were quantified by examining mRNA expression. The results showed that NHBE cells differentiated on the 80-layer PCL mesh exhibited higher mRNA expression of the specific cell markers (forkhead box protein 1 (*FOXJ1*)): ciliated cell, secretoglobin family 1A member 1 (*SCGB1A1*): club cells, and cytokeratin 5 (*CK5*): basal cell) than those differentiated on the 6-layer PCL mesh (Figure 3a). In particular, the expression of mucin 5AC and mucin 5B (*MUC5AC* and *MUC5B*, goblet cells) mRNA was significantly higher in cells differentiated on the 80-layer PCL mesh than that in cells differentiated on the 6-layer PCL mesh during the ALI. Furthermore, the *MUC5AC* mRNA expression in differentiated NHBE cells cultured on the 80-layer PCL mesh was significantly higher than that in cells cultured on the 6-layer PCL mesh 14 days after the initiation of the ALI.

Differentiated cells (i.e., goblet, ciliated, club, and basal cells) were identified using immunofluorescence. In the epithelial layers generated from NHBE cells on the 6- and 80-layer PCL meshes, MUC5AC (goblet cells), acetylated-tubulin (Ac-tubulin, ciliated cells), and CK5 (basal cells) were detected (Figure 3b). At the protein level, MUC5AC, Ac-tubulin, CK5 (basal cells), and CC10 (club cells) increased in a time-dependent manner (Figure 3c). In particular, the MUC5AC and CK5 levels were greater in the differentiated NHBE cells differentiated on the 80-layer PCL mesh than in those differentiated on the 6-layer PCL mesh.

### 2.4. Hyperplastic Goblet Cells on the 80-Layer PCL Mesh Compared to on the 6-Layer PCL Mesh

We next measured the mucin levels in the NHBE cells differentiated on the 6- and 80-layer PCL meshes. The results showed that the stained mucin levels of the differentiated NHBE cells on the 6- and 80-layer PCL mesh were slightly increased in a time-dependent manner during the ALI (Figure 4a). In particular, the stained mucin levels of the NHBE cells differentiated on the 80-layer PCL mesh were greater than those on the 6-layer PCL mesh from seven days post-initiation of the ALI (Figure 4a).

To identify the distribution of the differentiated cells, we detected specific markers (MUC5AC, Ac-tubulin, CK5, and CC10) using whole-mount immunostaining. The results confirmed that hyperplastic goblet cells (MUC5AC positive) were present on the 80-layer PCL mesh on day 21 after the initiation of the ALI (Figure 4b, Appendix A). Furthermore, ciliated cells (Ac-tubulin positive) were detected on the 6-layer PCL mesh but not on the 80-layer PCL mesh (Figure 4b). With the control, goblet cells and ciliated cells were detected on the PET Transwell (Appendix A).

### 2.5. The Expression of EMT Markers in Cells Differentiated on the 6- and 80-Layer PCL Meshes

To investigate the differences in the signaling pathway of NHBE cells differentiated on the 6- and 80-layer PCL meshes, the mRNA expression levels of EMT-related genes were quantified. The results showed that twist family BHLH transcription factor 1 (*TWIST*), *N-cadherin*, and snail family transcription repressor 1 (*SNAIL*) mRNA expression significantly increased in differentiated NHBE cells on both PCL meshes in a time-dependent manner (Figure 4c). In particular, the fold change in *N-cadherin* and *TWIST* expression in the differentiated NHBE cells on the 80-layer PCL mesh was greater than in those on the 6-layer PCL mesh on days 14 and 21 post-initiation of the ALI. Moreover, the expression of fibrosis-related transforming growth factor (*TGF*)-*β*, alpha smooth muscle actin (*α-SMA*), and *fibronectin* mRNA was increased more in NHBE cells differentiated on the 80-layer mesh than those on the 6-layer mesh (Figure 4d). To confirm the association with inflammation, mRNA expression of inflammation-related genes was quantified. Tumor necrosis factor (*TNF*)-*α* mRNA expression in NHBE cells differentiated on the 80-layer PCL mesh significantly increased from seven days post-initiation of the ALI (Figure 4e). In addition, interleukin (*IL*)-*6* mRNA expression increased in NHBE cells differentiated on the 80-layer PCL mesh from seven days post-initiation of the ALI. 

### 2.6. Oxidative Stress-Related Signaling Pathway in NHBE Cells Differentiated on the 80-Layer PCL Mesh

To investigate the involvement of oxidative stress, the expression of RAC family small GTPase 1 (RAC1)-nicotinamide adenine dinucleotide phosphate (NADPH) oxidase (NOX) was measured. *RAC1*, *NOX1*, *NOX2*, and *NOX4* mRNA expression increased in the NHBE cells differentiated on the 6-and 80-layer PCL meshes (Figure 5a) from seven days post-initiation of the ALI. Also, protein expression levels of the RAC1 and NOX4 were significantly higher in the NHBE cells differentiated on the 80-layer PCL mesh than those on the 6-layer PCL mesh at days 14 and 21 post-initiation of the ALI (Figure 5b). Furthermore, the SOX9 and NF-κB levels were higher in the NHBE cells differentiated on the 80-layer PCL mesh than those on the 6-layer PCL mesh at days 14 and 21 post-initiation of the ALI. The immunofluorescence assay results showed that the NOX2 and NOX4 levels in the NHBE cells differentiated on the 80-layer PCL mesh were greater than those in NHBE cells differentiated on the 6-layer PCL mesh at day 21 (Figure 5c). In addition, activator protein 1 (AP-1)-positive cells were more distributed in the NHBE cells differentiated on the 80-layer PCL mesh than those in the cells differentiated on the 6-layer PCL mesh (Figure 5d).

## 3. Discussion

Our results show that the NHBE cells differentiated on 80-layer PCL mesh generate less normal respiratory epithelium than NHBE cells differentiated on 6-layer PCL mesh, resulting in higher goblet cell numbers. Moreover, EMT- and oxidative stress-related markers were increased in the NHBE cells differentiated on 80-layer PCL mesh during the ALI.

ALI cell culture models were needed to evaluate the toxicity of gases and aerosols when the exposure route is the respiratory track by inhalation [8,29]. With their relevant environment and human respiratory epithelium-like barrier properties, ALI cell culture models are a more relevant tool for determining the amount of gas or aerosols absorbed into the body than submerged two-dimensional (2D) culture models [30]. PET Transwell inserts are frequently used in ALI models due to their permeability and low cost. However, the findings of several studies suggested that cell survival and differentiation were affected by the topology of the cell culture membrane [31,32,33]. Moreover, cell adhesion and proliferation have been shown to be enhanced in electrospun nanofiber scaffolds [34,35]. In studies by Yim et al. and Lee at al., human stem cells showed greater differentiation to neural cells when cultured on patterned surfaces versus on unpatterned surfaces [31,32]. In our study, NHBE cell survival was higher on electrospun PCL mesh compared to PET Transwell inserts. It seems that the patterned surface with fibrous structure of the electrospun PCL mesh, like that of the BM in vivo, increased NHBE cell survival compared to the smooth surface of the PET Transwell inserts.

The disruption of tight junctions is associated with various lung diseases [36,37]. TEER is frequently used to determine the integrity of the epithelium or evaluate barrier function in polarized epithelial cells. Primary cells have a broad range of TEER values (300–800 Ω/cm^2^); this is because the TEER can be affected by the type of donor cell, passage of cells, or culture medium [9,38]. However, it is generally accepted that the epithelium has formed a well-differentiated layer with proper barrier function when the TEER is above 200 Ω/cm^2^ [39]. Though limited studies have directly evaluated and reported on the TEER values of the human bronchus or trachea epithelium, electrophysiological values for rabbit bronchial epithelium have been measured ex vivo and reported as 266 ± 97 Ω/cm^2^ [40]. In our study, the TEER values for the NHBE cells differentiated on the six-layer PCL mesh were between 250 and 300 Ω/cm^2^ during the entire 28 days of the ALI. These TEER values were similar to those of the NHBE cells differentiated on the 80-layer PCL mesh, even though the overall thickness of the newly generated epithelium with the 6-layer PCL mesh itself was considerably less than that of the new epithelium with the 80-layer PCL mesh. Thus, the use of either the 6-layer PCL mesh or the 80-layer PCL mesh did not seem to impact the integrity of the newly generated epithelium.

The normal human airway is covered with pseudostratified epithelium, which contains four cell types: 50–70% ciliated cells that have cilia for removing particles and pathogens, 25% goblet cells that generate mucus (MUC5AC and MUC5B), 11% club cells that contain club cell-specific 10 kDa (CC10) secretory protein [2,41], and a small number of basal stem cells [9]. During ALI, the NHBE cells on the 6- and 80-layer PCL meshes differentiated into four types of cells: ciliated cells, goblet cells, club cells, and basal cell. However, the NHBE cells differentiated on the 80-layer PCL mesh showed significantly higher expression of MUC5AC and MUC5B than the NHBE cells differentiated on the 6-layer PCL mesh.

Mucociliary clearance is the first-line defense mechanism in the respiratory tract [42]. Two mucins, MUC5AC and MUC5B, are the main components of the respiratory mucus, and changes in the amount or components of mucus induce abnormal mucociliary clearance, accumulation of airway mucus, and dysfunctional ciliary movement [43,44,45]. Moreover, various respiratory diseases such as COPD and asthma are accompanied by increased production of mucus or goblet cells, which can even induce mucous metaplasia. These changes in the respiratory mucus can induce airway obstruction [46]. In addition, the changes in MUC5B can lead to dysfunctional mucociliary clearance, which increases airway inflammation [47]. However, the exact mechanism by which MUC5B accumulation leads to lung fibrosis has not been fully elucidated. In our study, the differentiated NHBE cell layer on the 80-layer PCL mesh showed changes in mucin secretion and increased goblet cells, which are features also observed with thickened BM in people with asthma. Moreover, the expression of MUC5AC and MUC5B was affected by the thickness of the electrospun PCL mesh. Previous studies that evaluated goblet cell function using an in vitro airway model used IL-13 to induce goblet cell hyperplasia [48]. Although we did not treat NHBE cells in our study with IL-13, the 80-layer PCL mesh was found to induce increased expression of MUC5AC and MUC5B during ALI. Thus, our findings indicated that a change in topology induced goblet cell hyperplasia in the epithelium during the ALI.

Increased EMT has been observed in patients with respiratory diseases such as COPD and asthma. Also, epithelial leakage caused by the disruption of tight junctions and goblet hyperplasia could increase EMT [49]. Abnormal respiratory epithelial cells had loose tight junctions, which means that TGF-β1 infiltrated deeper into the epithelium and induced more EMT [50]. Moreover, in the fibrous foci of the IPF lung, the increased expression of TGF-β1 was accompanied by increased ECM components, such as collagen or fibronectin [51,52]. The epithelium in IPF lungs expresses TNF-α, which amplifies inflammation [53]. In our study, the expression of TNF-α was higher in NHBE cells differentiated on the 80-layer PCL mesh than that in NHBE cells differentiated on the 6-layer PCL mesh. Several studies suggested that the topology of the cell culture membrane may have affected TNF-α expression, even in the absence of immune cells. Furthermore, the expression of TGF-β1 and other EMT markers such as N-cadherin and TWIST was substantially higher in NHBE cells differentiated on the 80-layer PCL mesh than those differentiated on the 6-layer PCL mesh. Thus, our ALI model using electrospun PCL mesh showed that the expression of EMT-related markers was altered only by the thickness of the membrane in the absence of TGF-β treatment.

Genetic alteration of SOX9 involved in the regeneration of the lung results in changes in cell proliferation, differentiation, and ECM synthesis [54,55]. SOX9 expression is increased in mouse lung fibrosis induced by TGF-α. SOX9 is also involved in the expression of EMT-related markers, such as SNAIL, snail family transcription repressor (SLUG), and zinc finger E-box-binding homeobox 1 (ZEB1) [56], and in mucous metaplasia [57]. Jiang et al. showed that SOX9 expression was increased in mucous metaplasia of an airway epithelium. In our study, SOX9 expression in NHBE cells differentiated on the 80-layer PCL mesh was significantly increased compared to that in NHBE cells differentiated on the 6-layer PCL mesh. Therefore, the membrane thickness affected SOX9 expression, which is related to goblet cell hyperplasia and EMT.

Our findings also raised the question of how the changes in SOX9 expression related to the thickness of the cell culture membrane had an effect at the cellular level. Jin et al. suggested that SOX9 triggers the activation of NF-κB, which is involved in various cellular responses, including proliferation, inflammation, and apoptosis as a positive feedback signal [58]. Our study showed that NF-κB and SOX9 expression were increased in the NHBE cells differentiated on the 80-layer PCL mesh compared to those cultured on the 6-layer PCL mesh. Thus, in the NHBE cells differentiated on the thick PCL mesh, SOX9 activated NF-κB, which then triggered goblet cell hyperplasia.

Additionally, oxidative stress upregulated NF-κB, which boosts the expression of inflammatory cytokines (e.g., TNF), and these cytokines further enhance oxidative stress [59]. Increased RAC1 and NOX1 levels also lead to increased NF-κB. Therefore, there is a vicious cycle involving oxidative stress and NF-κB expression. In our study, both NF-κB and RAC1 expression were increased in NHBE cells differentiated on the 80-layer PCL mesh compared to cells on the 6-layer PCL mesh. NOX4 expression was also higher in cells differentiated on the 80-layer PCL mesh than that in cells differentiated on the 6-layer mesh at day 7 after the initiation of the ALI. Since RAC1 can increase both NOXs and NF-κB, and NOXs can increase NF-κB, it is possible that RAC1 enhanced NF-κB activity either directly or through NOXs in our study. Moreover, the increased RAC1 observed in the NHBE cells differentiated on the 80-layer PCL mesh may have led to an increase in NF-κB, which eventually increased SOX9 expression. The increased SOX9 in the NHBE cells differentiated on the 80-layer PCL mesh may have triggered the observed increase in EMT and goblet cell hyperplasia in these cells. Though our findings provide some insight into the signaling pathways involved in the differentiation of NHBE cells cultured on synthetic membranes, further studies in which RAC1 or SOX9 is inhibited are needed to elucidate the pathway.

## 4. Materials and Methods

### 4.1. Fabrication of Electrospun PCL Mesh

The electrospun fibers were deposited using an electrospinning device (NanoNC, Seoul, Korea) on a collector, which was a flat plate connected to the ground. To prepare the electrospinning solution for the fabrication of the PCL nanofibers, PCL (Mn80000, Sigma-Aldrich, Darmstadt, Germany) was dissolved in formic acid (99.0%, Samchun Chemicals, Seoul, Korea) at a concentration of 15.0% (*w*/*v*). Twenty-five-gauge needles were used for electrospinning, and the solution was injected at a rate of 1.0 mL/h. The distance between the syringe needle and the collector was 10.0 cm. The operating voltage was 25.0 kV.

### 4.2. Electrospun PCL Mesh Characterization and Tensile Strength Evaluation

The electrospun PCL meshes were observed by scanning electron microscopy (SEM) (su8010, Hitachi, Tokyo, Japan) to assess their morphology. Briefly, scaffolds were sputter-coated with platinum (Hummer^™^ 6.2, Anatech Ltd., Denver, CO, USA) to a thickness of 7–8 nm. Images were acquired at an accelerating voltage of 5 kV, with a working distance of 15 cm. The pore sizes of the mesh were calculated from the acquired SEM images. Specifically, after converting the obtained SEM image into a black-and-white image, the image was divided into a 6 × 5 matrix, and the pore size was measured randomly at a total of 30 points. The tensile strength and elastic modulus of the membranes were measured using a universal testing machine (EZ-SX; Shimadzu, Kyoto, Japan). A load cell of 10 N was used in the machine, and the clamped meshes were pulled at speed of 3 mm/min until breakage. However, tensile strength of PCL meshes as a wet condition was not evaluated.

### 4.3. Preperation of PCL Mesh and Expansion of Primary Airway Epithelial Cells

Electrospun PCL mesh sheets (with a diameter of 1 cm) were punched out and inserted into cell crowns (Sigma-Aldrich, St. Louis, MO, USA). After insertion, the electrospun PCL meshes were sterilized with 70% ethanol and 20% antimycotic–antibiotic solution (Welgene, Daegu, Korea) and dried on a clean bench overnight at room temperature (RT). Cryopreserved NHBE cells (CC-2540S, LONZA, Morristown, NJ, USA) were seeded at passage zero (P0) into 100 mm culture dishes in Pneumacult-Ex Plus growth medium (StemCell technologies, Vancouver, BC, Canada). When the NHBE cells reached 70–80% confluency, they were passaged using trypsin-EDTA (Welgene, Daegu, Korea).

### 4.4. Differentiation of NHBE Cells by ALI

NHBE cells (passage 4–5) were seeded at 1 × 10^5^ cells/cm^2^ in 24-well plates containing electrospun PCL mesh and PET Transwell inserts (#3470, Corning, Steuben County, NY, USA) coated with 0.1 mg/mL human collagen type I (Purecol I, Advanced Biomatrix, USA). Pneumacult-Ex plus medium (StemCell technologies, Vancouver, BC, Canada) with 0.096 µg/mL hydrocortisone (StemCell technologies, Vancouver, BC, Canada) was added to the apical and basolateral chambers, and the medium was changed every second day. The NHBE cells were cultured under submerged conditions for five days at 37 °C in 5% CO_2_. The ALI medium was prepared by mixing Pneumacult-ALI medium (StemCell technologies, Vancouver, BC, Canada) with 0.48 µg/mL hydrocortisone and 0.2% heparin (StemCell technologies, Vancouver, BC, Canada) according to the manufacturer’s instructions. When the cell reached 100% confluency, the medium was removed from the apical and basolateral chamber, and 700 µL of the ALI medium was added to the basolateral chamber. The medium was changed every second day, and the apical surfaces were washed with Dulbecco’s phosphate-buffered saline (DPBS) each week. The cells were cultured for three weeks before the various tests were performed. 

### 4.5. Cell Viability Assay

The cell viability assay was performed using the Cell Counting Kit-8 (CCK-8; Dojindo, Kumamoto, Japan) at 0, 7, 14, 21, and 28 days after introducing the air-lift condition, following the manufacturer’s protocol. Briefly, the electrospun PCL mesh and PET Transwell inserts were transferred to new 24-well plates. Then, 1 mL of ALI medium and 100 µL of CCK-8 solution were added to each well. After incubation for 1 h at 37 °C in 5% CO_2_, absorbance was measured at a wavelength of 450nm. The following equation was used to calculate the cell viability: $$Cell\;viability\;\%=\frac{\left(At-Ab\right)}{\left(At0-Ab0\right)}\times 100$$
where
At is the absorbance of NHBE cells on the membrane under ALI conditions;Ab is the absorbance of the membrane only under ALI conditions;At0 is the absorbance of NHBE cells on the membrane under ALI day 0;Ab0 is the absorbance of the membrane only under ALI day 0.

### 4.6. Trans-Epithelial Electrical Resistance (TEER)

The TEER of the NHBE cells cultured on PCL mesh was measured at seven-day intervals (up to 28 days) during ALI using the TEER measuring system (Kanto Chemical CO., Tokyo, Japan). Briefly, the medium was added in the apical chamber with 0.3 mL of ALI medium, and that was added in the basolateral chamber with 0.7 mL of ALI medium. The TEER of the insert is expressed as Ω × cm^2^.

### 4.7. Histological and Immunochemical Analysis

At a specific time point after cell culture was initiated on the PCL mesh (i.e., during the ALI cell culture phase), the PCL mesh was fixed in 4% paraformaldehyde (PFA), processed, and embedded in polyester wax. Sections (5 µm thick) were cut and stained with Harris hematoxylin and eosin (H&E) after rehydration. Immunohistochemistry was used to detect markers expressed in the NHBE cells differentiated on the PCL mesh. Samples were fixed in 4% PFA at RT for 10 min and then permeabilized with 0.2% Triton-X 100 (Sigma-Aldrich, St. Louis, MO, USA) for 10 min. Subsequently, the samples were blocked for 60 min with 2% bovine serum albumin (BSA) (Sigma-Aldrich, St. Louis, MO, USA) in phosphate-buffered saline (PBS). The samples were then incubated with primary antibodies against MUC5AC (#198294, 1:200, Abcam, UK), CC10 (#sc-365992, 1:200, Santacruz, Dallas, TX, USA), Ac-tubulin (#T7451, 1:200, Sigma-Aldrich, St. Louis, MO, USA), CK5 (#MA5-170571:200,Thermo Fisher, Waltham, MA, USA), RAC1 (1:200, Cell Biolabs Inc., Sandiego, CA, USA), NOX2 (#MA5-18052, 1:200, Thermo Fisher, Waltham, MA, USA), and NOX4 (#ab154244, 1:200, Abcam, UK) overnight at 4 °C. Goat anti-mouse Alexa Flour 488 and 594 (#A11001 and #A11005, 1:400, Thermo Fisher, Waltham, MA, USA) and goat anti-rabbit Alexa Flour 488 and 594 (#A11008 and #A11012, 1:400, Thermo Fisher, Waltham, MA, USA) were used as secondary antibodies and added for 1 h at RT. The samples were then washed with DPBS, and 1 µg/mL DAPI (#D9542, Sigma-Aldrich, St. Louis, MO, USA) was added to counterstain the nuclei. The samples were placed on microscope slides (between the glass and coverslip) and observed using confocal laser scanning microscopy (LSM 700, Carl Zeiss, Oberkochen, Germany).

### 4.8. Alcian Blue Staining

Rehydrated sections were stained using an Alcian Blue Stain Kit (#ab150662, Abcam, UK) according to the manufacturer’s protocol.

### 4.9. Whole-Mount Immunostaining

Fixed NHBE cells differentiated on PCL mesh during the ALI were blocked with 2% BSA at 4 °C overnight. The cells were then incubated with primary antibodies against MUC5AC (#198294, 1:200, Abcam, UK), CC10 (#sc-365992, 1:200, Santacruz, Dallas, TX, USA), Ac-tubulin (#T7451, 1:200, Sigma-Aldrich, St. Louis, MO, USA), CK5 (1:200, Thermo Fisher, USA), RAC1 (1:200, Cell Biolabs Inc., Sandiego, CA, USA), NOX2 (#MA5-18052, 1:200, Thermo Fisher, USA), NOX4 (#ab154244, 1:200, Abcam, UK), and activator protein 1 (AP-1, #A5968, 1:500, Sigma-Aldrich, St. Louis, MO, USA) in 1% BSA overnight at 4 °C. The cells were washed with DPBS and incubated with secondary antibodies [goat anti-mouse Alexa Flour 488 and 594 (#A11001 and #A11005, 1:400, Thermo Fisher, Waltham, MA, USA) and goat anti-rabbit Alexa Flour 488 and 594 (#A11008 and #A11012, 1:400, Thermo Fisher, Waltham, MA, USA)] for 2 h at RT. Nuclei were stained with glycerol. Microscopic images were acquired using a Carl Zeiss LSM 700 microscope (20×) (LSM700, Carl Zeiss, Germany) and processed using ZEN software V3.4 (Zen Blue edition, Carl Zeiss, Germany).

### 4.10. Western Blot Analysis

Total cell lysates were isolated from the PCL mesh for western blotting. Briefly, PCL meshes with differentiated NHBE cells were added to RIPA lysis buffer with protease and phosphatase inhibitor. After centrifugation of lysates, supernatant was transferred to a new 1.5 mL tube, and protein concentration was measured using a Bradford assay. Equal concentrations (5 µg) of total protein were resolved on 10% polyacrylamide gels and electro transferred to polyvinylidene difluoride (PVDF) membranes. After the PVDF membranes were blocked using 5% skim milk, they were probed overnight at 4 °C with primary antibodies against the following proteins: MUC5AC (#198294, 1:1000, Abcam, UK), CC10(#sc-365992, 1:1000, Santacruz, Dallas, TX, USA), Ac-tubulin (#T7451, 1:1000, Sigma-Aldrich, St. Louis, MO, USA), CK5 (#MA5-17057, 1:1000, Thermo Fisher, Waltham, MA, USA), SOX9 (#ab185966, 1:1000, Abcam, UK), NF-κB (#8242, 1:1000, Cell Signaling, UK), RAC1 (1:200, Cell Biolabs Inc., Sandiego, CA, USA), NOX2 (#MA5-18052, 1:200, Thermo Fisher, Waltham, MA, USA), NOX4 (#ab154244, 1:200, Abcam, UK), and GAPDH (#sc-47778, 1:1000, Santacruz, Dallas, TX, USA). After washing with buffer, the membranes were incubated with appropriate secondary antibody for 2 h at RT.

### 4.11. Quantitative Real-Time Polymerase Chain Reaction (PCR)

Total RNA was isolated from the NHBE cells differentiated on PCL mesh using the ReliaPrep™ RNA Cell Miniprep system (Promega, Madison, WI, USA). Briefly, PCL meshes with differentiated NHBE cells were added to 0.5 mL TRIzol. After preparation of RNA, RNA was converted to cDNA using the RevertAid H Minus First Strand cDNA synthesis kit (Thermo Fisher, Waltham, MA, USA). Quantitative polymerase chain reactions (PCRs) were performed by mixing cDNA with SYBR Premix Taq (TaKaRa Bio Inc., Shiga, Japan) in a CFX96™ real-time system (BioRad, Hercules, CA, USA). Data were analyzed using the delta-delta CT method, and the expression level of each target gene was normalized to the expression level of GAPDH. The Appendix A list the primers used in the quantitative real-time PCR.

### 4.12. Statistical Analysis

All experiments were performed three times, and the average values are presented unless otherwise stated. Data are represented as the mean ± standard deviation. The statistical significance of the results was determined using Prism 5 software (GraphPad, San Diego, USA) and the Mann–Whitney U test. Significance was defined as * *p* < 0.05 and ** *p* < 0.01 for all experiments.

## 5. Conclusions

The aim of this study was to determine whether the thickness of the electrospun PCL mesh used to form cell culture membranes influenced the differentiation of airway epithelial cells. It was found that cell proliferation increased when a 6-layer or an 80-layer PCL mesh was used, and that the use of the 80-layer PCL mesh (vs. the 6-layer PCL mesh) led to more goblet cell hyperplasia and EMT in airway epithelial cells in an ALI cell culture model. Taken together, the findings of this study suggest that the thickness of the electrospun PCL mesh used to form cell culture membranes modulates the differentiation of airway epithelial cells as shown below (Figure 6).

## Figures and Tables

**Figure 1 ijms-25-06650-f001:**
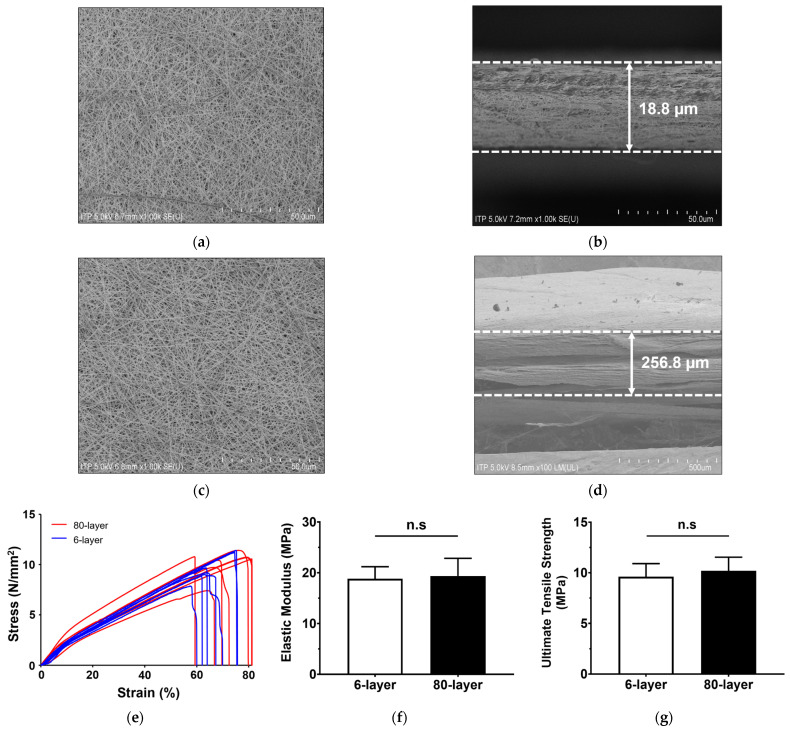
Characterization of electrospun PCL meshes. Characterization of the 6- and 80-layer electrospun PCL meshes. (**a**) Surface morphology of 6-layer PCL mesh. (**b**) Cross-sectional image of 6-layer PCL mesh. (**c**) Surface morphology of 80-layer PCL mesh. (**d**) Cross-sectional image of 80-layer PCL mesh. (**e**) Stress–strain diagrams of the two different PCL meshes. (**f**) Comparison of elastic modulus of the two different PCL meshes. (**g**) Comparison of the ultimate tensile strengths of the two different PCL meshes. “n.s” indicates not significant (*p* > 0.05).

**Figure 2 ijms-25-06650-f002:**
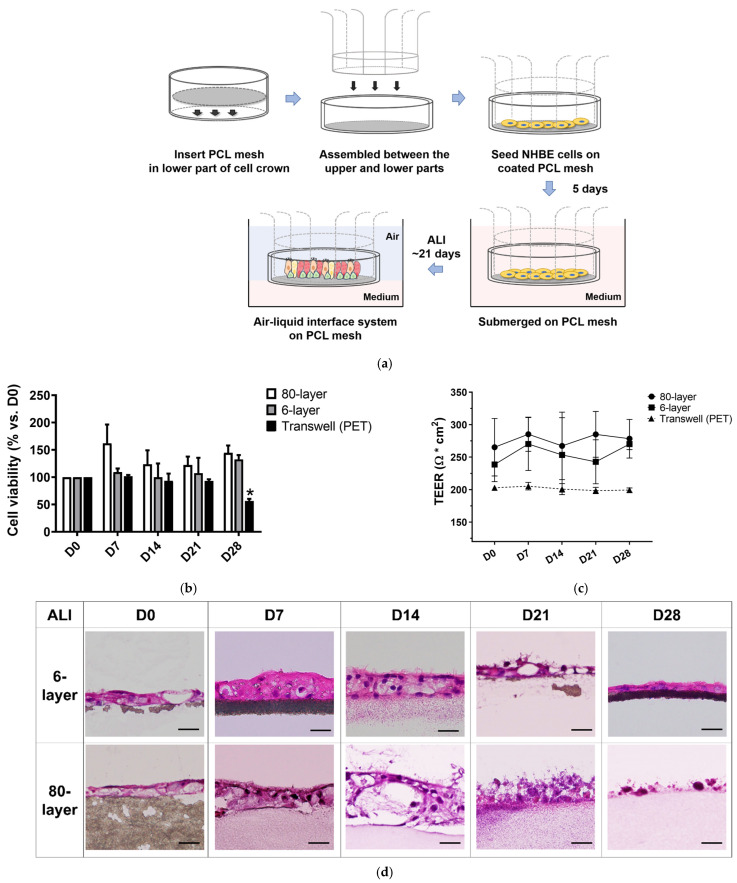
Airway epithelium on PCL mesh. (**a**) Scheme for the growth and differentiation of NHBE cells on the PCL mesh insert. (**b**) Viability of NHBE cells during the ALI on the 6- and 80-layer PCL meshes and PET Transwell inserts. Significance is denoted as * *p* < 0.05. (**c**) TEER values of NHBE cells differentiated on 6- and 80-layer PCL meshes. TEER (Ω × cm^2^) was quantified during the ALI. (**d**) H&E staining images of sectioned airway epithelium for 28 days (ALI) on 6- and 80-layer PCL meshes. Objective: 40×; scale bars: 20 μm.

**Figure 3 ijms-25-06650-f003:**
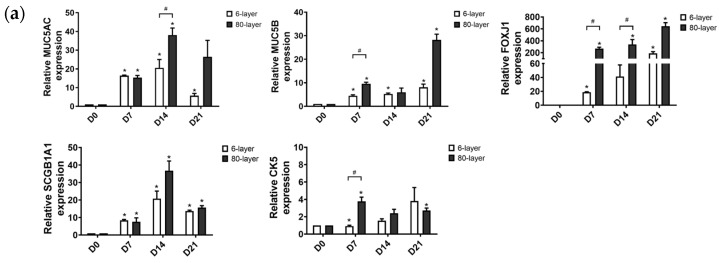
Establishment of normal and diseased airway epithelium on the 6- and 80-layer PCL meshes. (**a**) mRNA expression of specific cell markers in differentiated NHBE cells cultured on 6- and 80-layer PCL meshes. All data shown were obtained from at least three independent biological experiments. (**b**) Immunofluorescent images of sectioned NHBE cells differentiated on 6- and 80-layer PCL meshes for staining ciliated cells (Ac-tubulin, green), goblet cells (MUC5AC, red), club cells (CC10, green), basal cells (CK5, green), and nuclear stain (DAPI, blue). Scale bar: 20 μm. (**c**) Protein expression of specific cell markers in differentiated NHBE cells differentiated on PCL meshes. Each protein was normalized with GAPDH. All data shown were obtained from at least three independent biological experiments. Significance is denoted as * *p* < 0.05 in the differentiated NHBE cells at days 7, 14, and 21 compared to day 0 and as # *p* < 0.05 in the differentiated NHBE cells between the 6- and 80-layer PCL meshes at each time point.

**Figure 4 ijms-25-06650-f004:**
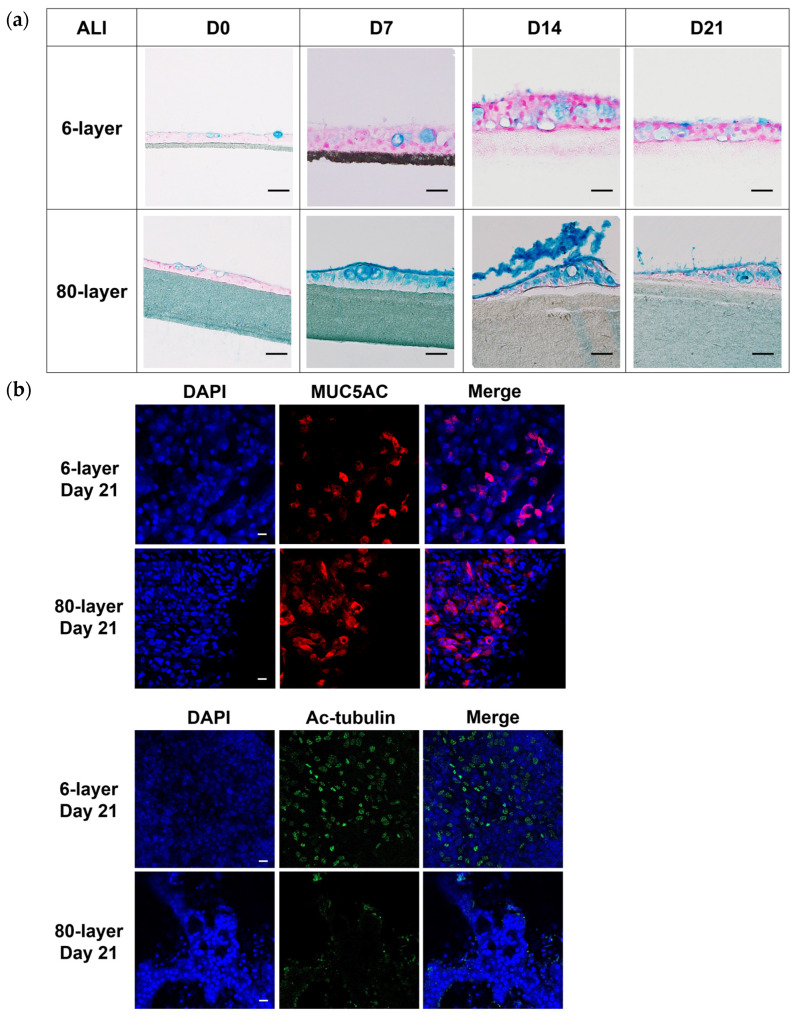
Goblet cell hyperplasia observed in NHBE cells differentiated on the 6- and 80-layer PCL meshes. (**a**) Stained mucin in NHBE cells differentiated on the 6- and 80-layer PCL meshes. Objective: 40×; scale bar: 20 μm. (**b**) Goblet cell and ciliated cell distributions on the 6- and 80-layer PCL meshes. Scale bars: 20 μm. (**c**) EMT-related markers in differentiated NHBE cells cultured on 6- and 80-layer PCL meshes. (**d**) Fibrosis-related markers in differentiated NHBE cells cultured on 6- and 80-layer PCL meshes. (**e**) Inflammation-related markers in differentiated NHBE cells cultured on 6- and 80-layer PCL meshes. Significance is denoted as * *p* < 0.05 in the differentiated NHBE cells at days 7, 14, and 21 compared to day 0 and as # *p* < 0.05 in the differentiated NHBE cells between the 6- and 80-layer PCL meshes at each time point. All the data were obtained from at least three independent biological experiments.

**Figure 5 ijms-25-06650-f005:**
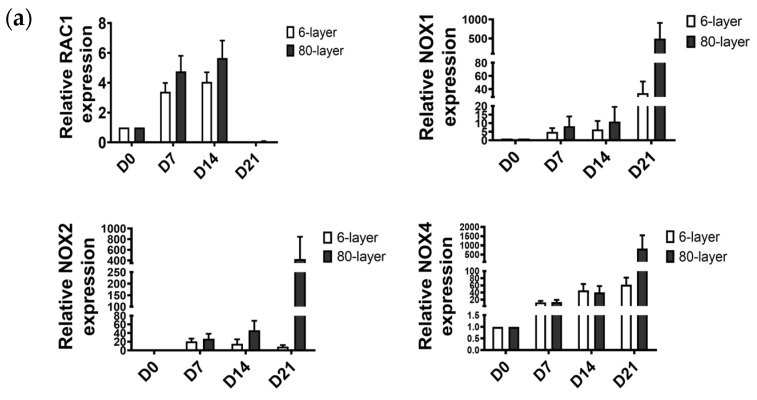
Expression of oxidative stress-related markers in the NHBE cells differentiated on PCL meshes. (**a**) Oxidative stress-related markers in the NHBE cells differentiated on 6- and 80-layer PCL meshes. (**b**) Protein expression of SOX9, NF-κB, RAC1, NOX2, and NOX4. Protein was quantified using western blotting. Each protein was normalized with GAPDH. (**c**) Immunofluorescence assay for oxidative stress-related markers in the NHBE cells differentiated on 6- and 80-layer PCL meshes at day 21 post-initiation of the ALI. RAC1 (green), NOX2 (green), NOX4 (red), and nuclear stain (DAPI, blue). (**d**) Whole-mount staining for AP-1 in cells differentiated on 6- and 80-layer PCL meshes. Scale bars: 20 µm. Significance is denoted as * *p* < 0.05 in the differentiated NHBE cells at days 7, 14, and 21 compared to day 0 and as # *p* < 0.05 in the differentiated NHBE cells between the 6- and 80-layer PCL meshes at each time point. All the data were obtained from at least three independent biological experiments.

**Figure 6 ijms-25-06650-f006:**
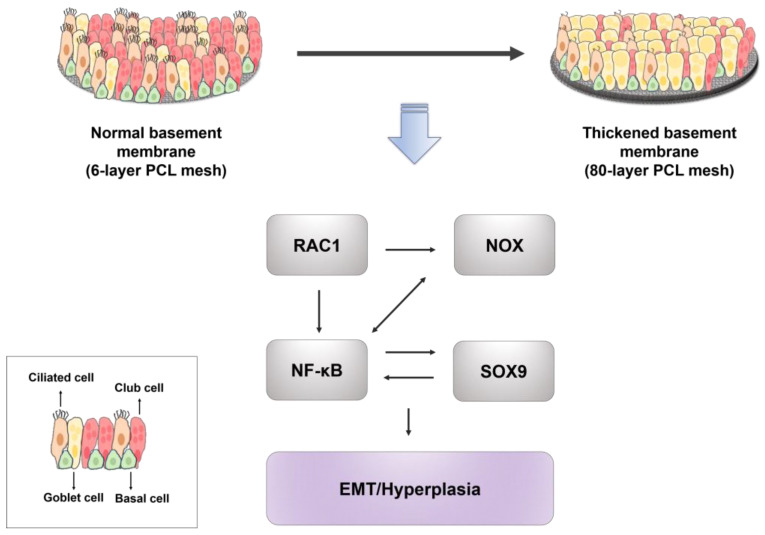
The signaling pathway for goblet-cell hyperplasia of NHBE cells on PCL meshes.

**Table 1 ijms-25-06650-t001:** Properties of electrospun PCL nanofibers.

Unit: µm		
	Mean ± SD (Min, Max)	
	6-Layer	80-Layer
Fiber diameter	0.23 ± 0.04 (0.18, 0.39)	0.25 ± 0.02 (0.20, 0.28)
Pore size	0.59 ± 0.22 (0.27, 1.15)	0.64 ± 0.41 (0.17, 1.81)
Membrane thickness	18.75 ± 1.92 (16.83, 20.67)	256.83 ± 21.64 ** (235.19, 278.47)

**, *p* < 0.01 vs. 6-layer.

## Data Availability

The data that support the findings of this study are available from the corresponding author upon reasonable request.

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
