# Peer review of "The Modulation of Respiratory Epithelial Cell Differentiation by the Thickness of an Electrospun Poly-ε-Carprolactone Mesh Mimicking the Basement Membrane"

_ijms, 2024, doi:10.3390/ijms25126650_

Round 1
Reviewer 1 Report
Comments and Suggestions for Authors
In the present study, Hui Son and et. al; evaluated the role of the basement membrane modulating bronchial cells differentiation. Specifically, they evaluate differences in basement membrane topology by using different scaffold thickness. Furthermore, they found that a thick scaffold affects the bronchial cell differentiation capacity, promoting hyperplasia, potentially mediated by NOX pathway. Although this study provides with potential mechanisms of bronchial cell differentiation, there are some concerns that need to be addressed.
Comments:
The authors might consider adding a PET control for the differentiation studies and not just for viability.
Further evaluation of using NOX inhibitors or ROS scavengers might need required to support the proposed mechanism. Does the hyperplastic phenotype revert? Or decreased in the presence of such compounds?
How is this study addressing Morris et al (PMID: 26275100); concerns regarding concentration of the polymer used, the rate at which the polymer solution was electrospun, and the needle’s diameter.
Due to the importance of reproducibility, the authors need to provide with a detail methodology of how they collect the cells for lysates for WB and RNA.
Please correct line 517, RRNA to RNA.
Reviewer 2 Report
Comments and Suggestions for Authors
as attached

Comments on the Quality of English Languageminor editing needed
Round 2
Reviewer 2 Report
Comments and Suggestions for Authors
according to the attachment.
